# INTERACTIVE VISUALIZATION FOR DEBUGGING RL

## ABSTRACT

Visualization tools for *supervised learning* (SL) allow users to interpret, intro-
spect, and gain an intuition for the successes and failures of their models. While
*reinforcement learning* (RL) practitioners ask many of the same questions while
debugging agent policies, existing tools aren't a great fit for the RL setting as these
tools address challenges typically found in the SL regime. Whereas SL involves
a static dataset, RL often entails collecting new data in challenging environments
with partial observability, stochasticity, and non-stationary data distributions. This
necessitates the creation of alternate visual interfaces to help us better understand
agent policies trained using RL. In this work, we design and implement an interac-
tive visualization tool for debugging and interpreting RL. Our system[1] identifies
and addresses important aspects missing from existing tools such as (1) visual-
izing alternate state representations (different from those seen by the agent) that
researchers could use while debugging RL policies; (2) interactive interfaces tai-
lored to metadata stored while training RL agents (3) a conducive workflow de-
signed around RL policy debugging. We provide an example workflow of how
this system could be used, along with ideas for future extensions.

## 1 INTRODUCTION

Machine learning systems have made impressive advances due to their ability to learn high dimen-
sional models from large amounts of data (LeCun et al., 2015). However, high dimensional models
are hard to understand and trust (Doshi-Velez & Kim, 2017). Many tools exist for addressing this
challenge in the *supervised learning* setting, which find usage in tracking metrics (Abadi et al., 2015;
Satyanarayan et al., 2017), generating graphs of model internals (Wongsuphasawat et al., 2018), and
visualizing embeddings (van der Maaten & Hinton, 2008). However, there is no corresponding set
of tools for the reinforcement learning setting. At first glance, it appears we may repurpose exist-
ing tools for this task. However, we quickly run into limitations, that arise due to the intent with
which these tools were designed. Reinforcement learning (RL) is a more interactive science (Neftci
& Averbeck, 2019) compared to supervised learning, due to a stronger feedback loop between the
researcher and the agent. Whereas supervised learning involves a static dataset, RL often entails col-
lecting new data. To fully understand an RL algorithm, we must understand the effect it has on the
data collected. Note that in supervised learning, the learned model has no effect on a fixed dataset.

Visualization systems are important for overcoming these challenges. At their core visualization
systems, consist of two components: *representation* and *interaction*. *Representation* is concerned
with how data is mapped to a representation and then rendered. *Interaction* is concerned with the
dialog between the user and the system as the user explores the data to uncover insights (Yi et al.,
2007). Though appearing to be disparate, these two processes have a symbiotic influence on each
other. The tools we use for representation affect how we interact with the system, and our interaction
affects the representations that we create. Thus, while designing visualization systems, it is impor-
tant to think about the application domain from which the data originates, in this case, reinforcement
learning.

Using existing tools we can plot descriptive metrics such as cumulative reward, TD-error, and action
values, to name a few. However, it is harder to pose and easily answer questions such as:

- How does the agent state-visitation distribution change as training progresses?
- What effect do noteworthy, influential states have on the policy?

---

[1] An interactive (anonymized) demo of the system can be found at `https://vizarel-demo.`
`herokuapp.com`

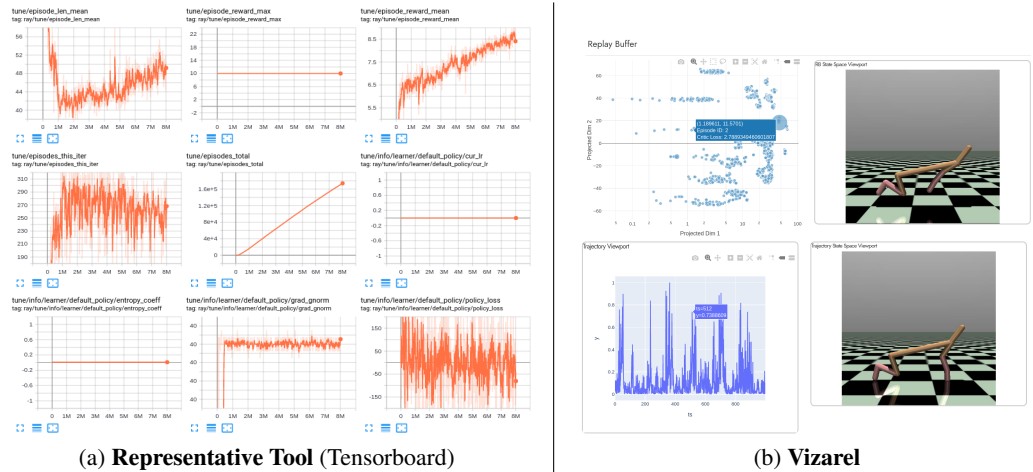

(a) **Representative Tool** (Tensorboard)    (b) **Vizarel**

Figure 1: **Tool Comparison** Contrasting between a representative tool for debugging RL in the existing ecosystem (**L**), and Vizarel (**R**), highlights the difference in design intent between both systems. The former was designed for the *supervised learning* setting, and has shown promise for use in *reinforcement learning*. However, we argue that there exists a large space of unexplored interfaces that could help aid the process of debugging RL algorithms and trained policies. We explore one such solution that is designed around the spatial and temporal aspects of training RL agents. This approach might help increase understanding, interpretability, and thereby serve as a complement to tools in the existing ecosystem.

- Are there repetitive patterns across space and time that result in the observed agent behavior?

These are far from an exhaustive list of questions that a researcher may pose while training agent policies, but are chosen to illustrate the limitations created by our current set of tools that prevent us from being able to easily answer such questions.

This paper describes our attempt at constructing Vizarel [2], an interactive visualization system to help interpret RL algorithms, debug RL policies, and help RL researchers pose and answer questions of this nature. Towards these goals, we identify features that an interactive system for interpretable reinforcement learning should encapsulate and build a prototype of these ideas. We complement this by providing a walkthrough example of how this system could fit into the RL debugging workflow and be used in a real scenario to debug a policy.

## 2    RELATED WORK

As we have argued in the introduction, existing visualization tools for machine learning primarily focus on the supervised learning setting. However, the process of designing and debugging RL algorithms requires a different set of tools, that can complement the strengths and overcome the weaknesses of offerings in the current ecosystem. In the rest of this section, we highlight aspects of prior work upon which our system builds. To the best of our knowledge, there do not exist visualization systems built for interpretable reinforcement learning that effectively address the broader goals we have identified. There exists prior work, aspects of which are relevant to features which the current system encapsulates, that we now detail.

**Visual Interpretability** Related work for increasing understanding in machine learning models using visual explanations includes: feature visualization in neural networks (Olah et al., 2017; Simonyan et al., 2014; Zeiler & Fergus, 2013), visual analysis tools for variants of machine learning models (Strobelt et al., 2017; Kahng et al., 2017; Kapoor et al., 2010; Krause et al., 2016; Yosinski et al.), treating existing methods as composable building blocks for user interfaces (Olah et al., 2018), and visualization techniques for increasing explainability in reinforcement learning (Rupprecht et al., 2020; Atrey et al., 2020; McGregor et al., 2015)

**Explaining agent behavior** There exists related work that tries to explain agent behavior. Amir & Amir summarize agent behavior by displaying important trajectories. van der Waa et al. (2018) introduce a method to provide contrastive explanations between user derived and agent learned policies. Huang et al. (2017) show maximally informative examples to guide the user towards understanding

---

[2]Vizarel is a portmanteau of visualization + reinforcement learning.

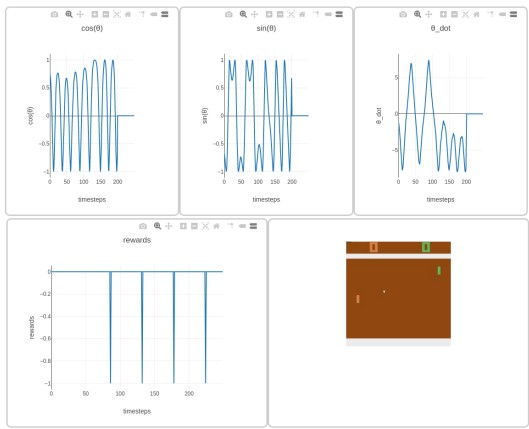

Figure 2: **State + Action Viewports (T)** Visualizing the state viewport for the inverted pendulum task. This representation overlayed with another state viewport similar to (b), provides the user with better intuition about the correspondence between states and actions for non image state spaces. **(B)** Visualizing the action viewport for the Pong environment (Bellemare et al., 2013). Hovering over instantaneous timesteps dynamically updates the state viewport (4.1.1) and shows the corresponding rendered image for the selected state. This representation provides the user with intuition about the agent policy, and could help subsequent debugging.

the agent objective function. Hayes & Shah (2017) present algorithms and a system for robots to synthesize policy descriptions and respond to human queries.

**Explainable reinforcement learning** Puiutta & Veith (2020) provide a survey of techniques for explainable reinforcement learning. Related work in this theme includes Puri et al. (2020); Reddy et al. (2019); Calvaresi et al. (2019); Juozapaitis et al. (2019); Sequeira & Gervasio (2020); Fukuchi et al. (2017); Madumal et al. (2020)

Similar to Amir & Amir; van der Waa et al. (2018); Huang et al. (2017), this work is motivated by the aim to provide the researcher relevant information to explore a possible space of solutions while debugging the policy. Similar to Hayes & Shah (2017), we present a functioning system that can respond to human queries to provide explanations. However, in contrast, the interactive system we present is built around the RL training workflow, and designed to evolve beyond the explanatory use case to complement the existing ecosystem of tools (Abadi et al., 2015; Satyanarayan et al., 2017). In contrast to the techniques surveyed in Puiutta & Veith (2020), the contribution here is not on any single technique to increase interpretability, but a whole suite of visualizations built on an extensible platform to help researchers better design and debug RL agent policies for their task.

## 3 PRELIMINARIES

We use the standard reinforcement learning setup (Sutton & Barto, 2018). An agent interacting with an environment at discrete timesteps $t$, receiving a scalar reward $r(s_t, a_t) \in \mathbb{R}$. The agent's behavior is defined by a policy $\pi$, which maps states $s \in S$, to a probability distribution over actions, $\pi : S \to P(A)$. The environment can be stochastic, which is modeled by a Markov decision process with a state space $S$, action space $A \in R^n$, an initial state distribution $p(s_0)$, a transition function $p(s_{t+1} \mid s_t, a)$, and a reward function $r(s_t, a_t)$. The future discounted return from a state $s_t$ and action $a_t$ is defined as $R_t = \sum_{i=t}^{T} \gamma^{i-t} r(s_t, a_t)$, with a discount factor $\gamma \in [0, 1]$. We use a replay buffer (Mnih et al., 2013; Lin) to store the agent's experiences $e_t = (s_t, a_t, r_t, s_{t+1})$ in a buffer $B = \{e_0, e_1, ..., e_T\}$.

## 4 VIZAREL: A TOOL FOR INTERACTIVE VISUALIZATION OF RL

This section describes how our interactive visualization system (Vizarel), is currently designed. The system offers different views that allow the user to analyze agent policies along spatial and temporal dimensions (described later in further detail). The tool consists of a set of *viewports*, that provide the user with different representations of the data, contingent on the underlying data stream. Viewports are generated by chaining together different visualization elements, such as:

1. *image buffers*: visualize observation spaces (image and non-image based)

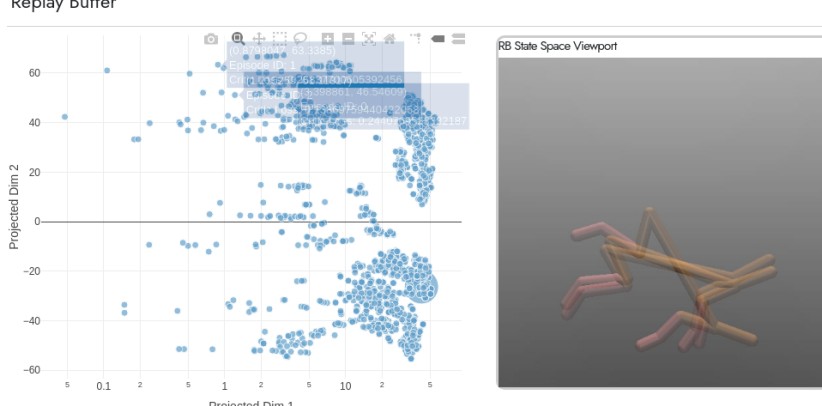

Figure 3: **Replay Buffer Viewport** Projecting the contents of the replay buffer into a 2D space for easier navigation, and clustering states based on similarity. This viewport provides a visual representation for replay buffer diversity and can help in subsequent debugging. Hovering over points in the replay buffer dynamically updates the generated state viewport (4.1.1), and shows the rendered image for the corresponding state (animation depicted using overlay).

2. *line plots*: visualize sequentially ordered data, such as action values or rewards across time
3. *scatter plots*: to visualize embedding spaces or compare tensors along specified dimensions
4. *histograms*: visualize frequency counts of specified tensors or probability distributions

The current implementation provides core viewports (detailed further), but can easily be extended by the user to generate additional viewports to explore different visualization ideas. This design naturally leads to the idea of an ecosystem of plugins that could be integrated into the core system, and distributed for use among a community of users to support different visualization schemes and algorithms. For example, the user could combine image buffers and line plots in novel ways to create a viewport to visualize the the state-action value function Sutton & Barto (2018). In the rest of this section, we provide details and distinguish between two types viewports currently implemented in Vizarel: temporal viewports and spatial viewports. Discussion on viewports beyond these has been deferred to the appendix. Comprehensive information on adding new viewports is beyond the scope of the paper, but has been described at length in the system documentation[3].

## 4.1 TEMPORAL VIEWS

Temporal views are oriented around visualizing the data stream (e.g. images, actions, rewards) as a sequence of events ordered along the time dimension. We have implemented three types of temporal viewports: state viewports, action viewports, and reward viewports, which we now detail.

### 4.1.1 STATE VIEWPORT

For visualization, we can classify states as either image-based or non-image based. The type of observation space influences the corresponding viewport used for visualization. We provide two examples that illustrate how these differing observation spaces can result in different viewports. Consider a non-image based observation space, such as that for the inverted pendulum task. Here, the state vector $\vec{s} = \{\sin(\theta), \cos(\theta), \dot{\theta}\}$, where $\theta$ is the angle which the pendulum makes with the vertical.

We can visualize the state vector components individually, which provides insight into how states vary across episode timesteps (Figure 2). Since images are easier for humans to interpret, we can generate an additional viewport using image buffers, that tracks changes in state space to the corresponding changes in image space. Having this simultaneous visualization is useful since it now enables us to jump back and forth between the state representation which the agent receives, and the corresponding image representation, by simply hovering over the desired timestep in the state viewport.

---

[3]Vizarel is planned to be released as an open source tool

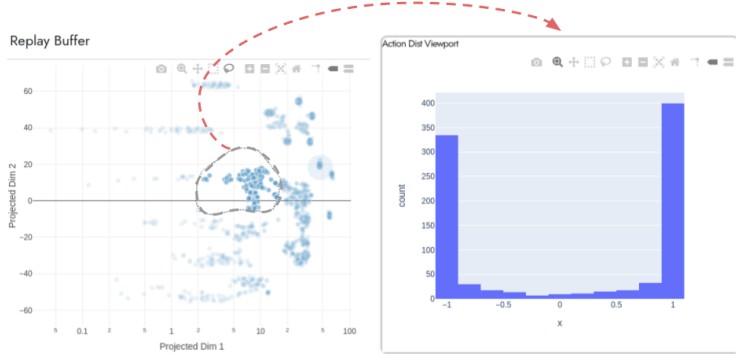

Figure 4: **Distribution Viewport** Using the lasso tool to select a group of points (dashed gray line) in the replay buffer viewport (4.2.1), dynamically updates (dashed red line) the distribution viewport (4.2.2) by computing and plotting the distribution of values for the specified tensor (e.g. actions or rewards).

For environments that have higher dimensional state spaces, such as that of a robotic arm with multiple degrees of freedom, we can visualize individual state components. However, since this may not be intuitive, we can also generate an additional viewport to display an image rendering of the environment to help increase interpretability.

### 4.1.2   ACTION VIEWPORT

The action viewport is used to visualize how the actions chosen by the agent vary across the episode (Figure 2). Consider the Pong environment (Bellemare et al., 2013), where the action $a$ at timestep $t$, corresponds to the direction in which the paddle should move. A visualization such as the one shown in Figure 2 can be generated to show the correspondence between actions and states. This allows the user to easily identify states marked by sudden action transitions, and thus aid debugging. This idea can easily be extended to agents with stochastic actions, where we could generate a viewport using histograms to visualize the change in action distribution over time.

For higher dimensional action spaces we can use a technique similar to the one used for state viewports, to generate multiple viewports that track individual action dimensions, for example joint torques for a multilink robot.

### 4.1.3   REWARD VIEWPORT

The reward viewport is used to visualize how the rewards received by an agent vary across the episode. A user can look at the reward viewport together with the state viewport to understand and find patterns across state transitions that result in high reward. For many environments, the reward function consists of components weighted by different coefficients. These individual components are often easier to interpret since they are usually correspond to a physically motivated quantity tied to agent behaviors that we wish to either reward or penalize. For example, in autonomous driving environments the reward can be formulated as a function of speed, collision penalties, and the distance from an optimal trajectory (Agarwal et al.).

In situations where we have access to these reward components, we can generate multiple viewports each of which visualize different components of the reward function. The viewports discussed so far can be combined to provide the user more insight into the correspondence between states (state viewport), actions (action viewport), and the components of the reward function (reward viewport) that the agent is attempting to maximize. Such a visualization could help alert researchers to reward hacking (Amodei et al., 2016).

### 4.2   SPATIAL VIEWS

Spatial views are oriented around visualizing the data stream as a spatial distribution of events. We have implemented three types of spatial viewports: replay buffer viewports, distribution viewports, and trajectory viewports, that we now describe.

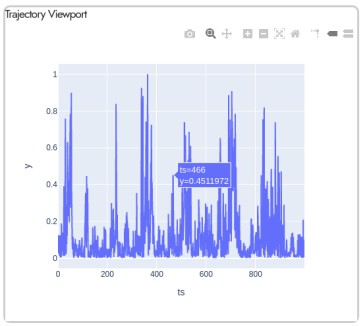 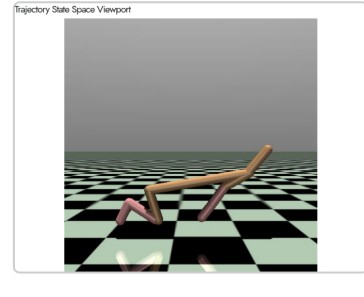

Figure 5: **Trajectory Viewport** Selecting points in the replay buffer viewport (4.2.1), causes the trajectory viewport (4.2.3) to dynamically update and plot the absolute normalized TD error values over the length of the trajectory. Hovering over points in the trajectory viewport, allows the user to view a rendering of the state corresponding to that timestep in the generated state viewport (4.1.1).

### 4.2.1 REPLAY BUFFER VIEWPORT

As formulated in §3, the replay buffer stores the agent's experiences $e_t = (s_t, a_t, r_t, s_{t+1})$ in a buffer $B = \{e_0, e_1, ..., e_T\} \forall i \in [0, T]$. For off-policy algorithms, the replay buffer is of crucial importance, since it effectively serves as an online dataset for agent policy updates (Fu et al., 2020). In the supervised learning setting, there exist tools[4] to visualize datasets, that provide the user with an intuition for the underlying data distribution. The replay buffer viewport aims to provide similar intuitions for the reinforcement learning setting by visualizing the distribution of data samples in the replay buffer.

Since the individual elements of the replay buffer are at least a four-dimensional vector $e_t$, this rules out the possibility of generating viewports to visualize data in the original space. We can instead visualize the data samples by transforming the points (van der Maaten & Hinton, 2008) to a lower-dimensional representation. This technique helps visualize the distribution of samples in the replay buffer, which is a visual representation of the replay buffer diversity (de Bruin et al.).

The size of the replay buffer can be quite large (Zhang & Sutton, 2018), which can lead to difficulties while navigating the space of points visualized in the replay buffer viewport. To nudge (Thaler & Sunstein, 2009) the user towards investigating samples of higher potential interest, we scale the size of points in proportion to the absolute normalized TD error (Sutton & Barto, 2018), which has been used in past work (Schaul et al., 2016) as a measure of sample priority during experience replay.

Moreover, the replay buffer viewport can be combined with the state viewport to simultaneously visualize an image rendering of the state, by tracking changes as the user hovers over points in the replay buffer viewport (Figure 3).

### 4.2.2 DISTRIBUTION VIEWPORT

The distribution viewport (Figure 4) complements the replay buffer viewport by allowing the user to select clusters of data samples and ask questions regarding the distribution of action, rewards, and other relevant tensors for the selected group of points.

Users might ask questions like:

- What is the distribution of actions the agent took for these groups of similar states?
- What is the distribution of rewards for the state action transitions?
- What is the overall diversity of states which the agent has visited?

If the updates to the agent policy result in better task performance, the entropy of the action distribution should reduce over time (discounting any external annealing caused due to exploration), which can be easily verified through this viewport. In the limit, the distribution of actions for a group of similar points should converge to a Dirac distribution, since the optimal policy for an infinite horizon discounted MDP is a stationary distribution (Puterman, 2005). In practice, observing the distribution converging around the mean value could indicate a promising policy training experiment.

---

[4]https://github.com/PAIR-code/facets

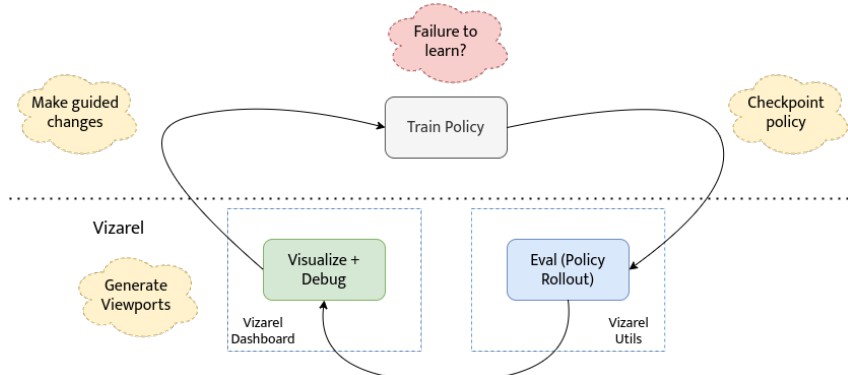

Figure 6: **Vizarel Workflow Diagram** Typical steps during policy debugging, and how the designed system fits into this workflow. The system takes as input a policy saved during a checkpoint and evaluates the policy through a specified number of rollouts. This data is then visualized through viewports specified by the user, that are used for debugging the policy through making guided changes.

For multi-dimensional action spaces, the viewport could be repurposed to display the variance of the action distribution, plot different projections of the action distribution, or use more sophisticated techniques such as projection pursuit (Huber, 1985).

### 4.2.3 TRAJECTORY VIEWPORT

A fusion of the components from the spatial and temporal views leads to a *spatio-temporal view*, an example of which is the trajectory viewport (Figure 5). The replay buffer viewport alone visualizes the spatial nature of the points in the replay buffer but does not display the temporal nature of trajectories. Being able to switch between spatial and temporal views is crucial when understanding and debugging policies. This is supported by selecting points in the replay buffer viewport, which then retrieves the corresponding trajectory.

The X coordinate in the trajectory viewport represents the timestep, and the Y coordinate is the absolute TD error, normalized to lie within $[0, 1]$. Hovering over points in the trajectory viewport retrieves an image rendering of the corresponding state in the state viewport. This correspondence enables the user to easily navigate through and visualize action sequences in the trajectory that consistently have a high TD error, thus speeding up debugging of policies.

## 5 WALKTHROUGH

We now detail an example workflow of how the system can be used in a real scenario. Figure 6, illustrates how Vizarel fits into an RL researcher's policy debugging workflow. Training a successful agent policy often requires multiple iterations of changing algorithm hyperparameters.

To speed up and increase the intuitiveness of this process, the researcher can load a stored checkpoint of the policy into the system, and evaluate a specified number of policy rollouts. Empirically, we've found that there should be enough rollouts to ensure sufficient coverage of the state space, since this influences the scope of questions which can be posed during debugging (e.g. through the replay buffer viewport). These rollouts can then be visualized and interacted with through specifying the required data streams and generating different viewports.

Figure 7, shows an example of replay buffer, state, and trajectory viewports generated for a

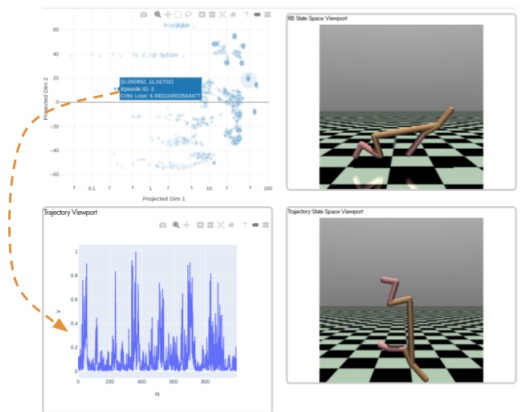

Figure 7: **Spatio-Temporal Interaction** Visualizing the replay buffer viewport (4.2.1) (spatial view), and trajectory viewport (4.2.3) (temporal view), along with overlays to independently track image renderings of states in both as a state space viewport (4.1.1). Navigating between these viewports allows the user to observe both agent spatial and temporal behavior, which could facilitate better insights during debugging.

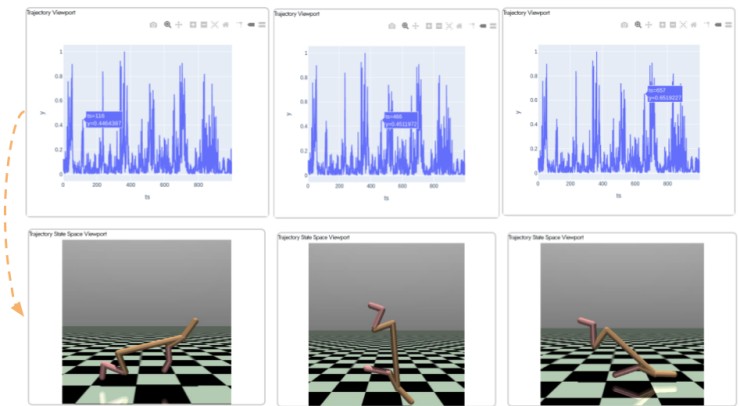

Figure 8: **Comparing TD error along an agent trajectory** Visualizing the trajectory viewport (4.2.3), allows the user to compare the TD error at different timesteps along the trajectory, along with the associated state viewport (4.1.1). An example interaction is visualized here by hovering over regions of potential interest in the trajectory viewport. This simultaneous view allows the user to easily compare and draw similarities between action sequences which cause large changes in TD error.

policy trained using DDPG on the HalfCheetah
task. The high variance in the TD error suggests the presence of critic overestimation bias (Thrun & Schwartz), which could be remedied by using algorithms known to reduce the impact of this issue (Fujimoto et al., 2018; Hasselt, 2010). Figure 8 shows how the user can compare the TD error along the agent trajectory. Hovering over regions of potential interest in the trajectory viewport allows the user to find action sequences that cause high variance in the TD error. A similar technique could be used to visualize clusters of states in the replay buffer space with high TD error (Figure 3). This approach could enable the user to identify patterns in states across space or time that persistently have high TD error, and design methods to mitigate this (Amodei et al., 2016).

Another approach the user could take is to generate a distribution viewport (Figure 4), and identify the distribution of actions in the vicinity of states with a high TD error. If similar states persistently have a higher action and/or reward variance, this suggests that the usage of variance reduction techniques could help learning (Schulman et al., 2018; Romoff et al., 2018). Once promising avenues for modification have been identified, the user can make guided changes, and retrain the policy.

## 6  CONCLUSION

In this paper, we have introduced a visualization tool, Vizarel, that helps interpret and debug RL algorithms. Existing tools which we use to gain insights into our agent policies and RL algorithms are constrained by design choices that were made for the supervised learning framework. To that end, we identified features that an interactive system for debugging and interpreting RL algorithms should encapsulate, built an instantiation of this system which we plan to release as an open source tool, and provided a walkthrough of an example workflow of how the system could be used.

There are multiple features under development that contribute towards both the core system. One feature is the integration of additional data streams such as saliency maps (Greydanus et al., 2018) to complement the state viewport. Another is designing the capability to use the system in domains that lack a visual component, for example in healthcare (Yu et al., 2020) or education (Reddy et al.). An extension is to add search capabilities that allow the user to easily traverse, query, and identify regions of interest in the replay buffer viewport.

Vizarel suggests a number of avenues for future research. First, we hypothesize that it could help design metrics that better capture priority during experience replay (Schaul et al., 2016). Second, it could help the researchers create safety mechanisms early on in the training process through identifying patterns in agent failure conditions (Amodei et al., 2016). Another possible research direction this tool catalyzes is the construction of reproducible visualizations through plugins integrated into the core system.

We anticipate that the best features yet to be built will emerge through iterative feedback, deployment, and usage in the broader reinforcement learning and interpretability research communities.

ACKNOWLEDGMENTS

SVD is supported by the CMU Argo AI Center for Autonomous Vehicle Research. BE is supported by the Fannie and John Hertz Foundation and the National Science Foundation (DGE1745016). Any opinions, findings, recommendations, and conclusions expressed in this material are those of the author(s) and do not reflect the views of funding agencies.

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

## APPENDIX

### ADDITIONAL VIEWPORT DETAILS

### TENSOR COMPARISON VIEWPORT

For environments that have higher dimensional action spaces, it is hard for the user to understand how neighboring points in the replay buffer viewport differ. This becomes especially relevant for diagnosing clusters of points that have a higher TD error. The tensor comparison viewport (Figure 9) enables the user to easily select points and then compare them along the dimensions of interest, which for example could be actions. Dimensions that have a standard deviation beyond a specified threshold are automatically highlighted, which enables the user to focus on the dimensions of interest.

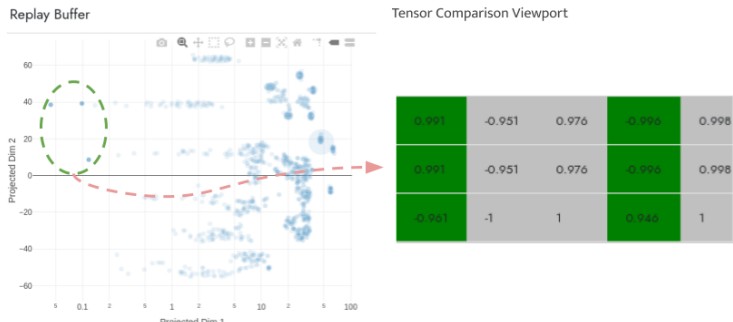

Figure 9: **Tensor Comparison Viewport** Selecting points (dashed green line) in the replay buffer viewport (4.2.1), and generating (dashed red line) the tensor comparison viewport, allows the user to compare tensors (e.g. actions or states), where dimensions of higher variance are automatically highlighted. This could lead to faster debugging in environments where each dimension corresponds to physically intuitive quantities.

### CONSTRUCTING VIEWPORTS

We describe how a user could create a new viewport through an example. However, we defer an extended discussion to the system documentation, since reading the source code and give the user more insight. As mentioned in Section 4, viewports are generated by chaining together different visualization elements. Various viewports we introduced along with the visualization elements they make use of are:

1. *image buffers*: state viewport
2. *line plots*: action viewport, reward viewport, trajectory viewport
3. *scatter plots*: replay buffer viewport
4. *histograms*: distribution viewport

Note that these primitives are not fixed and are bound to change if the creation of different viewports necessitates their expansion. However, we've found them to be a good starting point to provide the minimal functionality required to construct new viewports. For example the construction of a *saliency map viewport*, could be done using an *image buffer* and a *line plot*.

There exist utilities in the system to handle the rollout of agent policies, and storage of generated metadata. However, the user would still need to provide code to generate saliency maps [5]. Once the metadata has been generated, the user specifies which viewports to generate (e.g. core viewports and custom viewports) along with a visual layout for the dashboard. The system then generates an interactive interface with the specified viewports and layout, that the user can use to debug the agent policy and perform further analysis.

This viewport could further be incorporated as a plugin or extension to the core system and distributed to a community of users in the future. The exact details for this are not concrete yet, since we expect there to emerge a robust process through iterative design changes, as the tool finds broader usage.

---

[5]for which there exist open source tools

ALGORITHMS

REPLAY BUFFER VIEWPORT: PROJECTION

---

**Algorithm 1:** Procedure to generate samples for the replay buffer viewport

---

**Result:** Data samples from replay buffer transformed from original space $\mathbb{R}^n$ to lower
dimensional space $\mathbb{R}^d$, where $n >> d$

Load data samples $d_i$ from replay buffer into memory;
Where, $d_i = (s_i, a_i, r_i, s_{i+1})$, $i \in (0, T)$ and $T$ is the episode timestep;
**for** *i=0; i < T; i++* **do**
   |  *// create data matrix from replay buffer samples*;
   |  data[i] = $d_i$;
**end**
transform = compute_transform(data, type); *// where, type $\in$ [PCA, TSNE, UMAP]*
reduced_points = transform(data);

---

REPLAY BUFFER VIEWPORT: SAMPLE INFLUENCE VISUALIZATION

---

**Algorithm 2:** Procedure to compute visual size of data samples in the replay buffer viewport

---

**Result:** Visual size of data samples in replay buffer viewport transformed in proportion of their
influence on the agent policy

Load data samples $d_i$ from replay buffer into memory;
Where, $d_i = (s_i, a_i, r_i, s_{i+1})$, $i \in (0, T)$ and $T$ is the episode timestep;
Load sample metadata $m_i$ into memory;
Where, $m_i$ stores the TD error for each sample (for relevant algorithms) ;
**for** *i=0; i < T; i++* **do**
   |  data[i] = $m_i$;
**end**
d_min = min(data);
d_max = max(data);
**for** *i=0; i< T; i++* **do**
   |  *// p[i] stores the normalized TD error value, and (c_min, c_max) is the range to which*
   |   *values are mapped*;
   |  p[i] = $\frac{p[i] - d\_min}{d\_max - d\_min} \times (c\_max - c\_min) + (c\_min)$;
   |  *// r[i] stores the radius of the point to be plotted in the replay buffer viewport* ;
   |  r[i] = $\sqrt{\frac{p[i]}{\pi}}$;
**end**

---

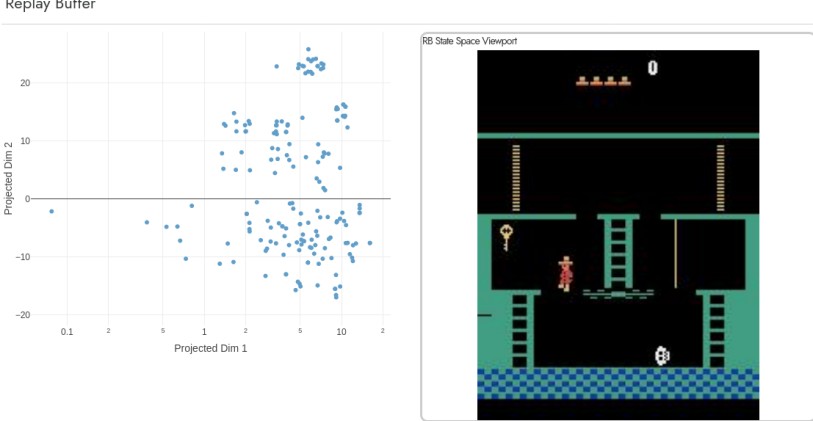

Figure 10: **Visualizing the replay buffer for hard exploration tasks** Tasks such as Montezuma's revenge are classic examples of hard exploration tasks. Here we show how the replay buffer viewport, can help visualize the distribution of data samples in the replay buffer.

## ALTERNATE ENVIRONMENTS

### HARD EXPLORATION

We've also experimented with using this tool for procedural, hard-exploration tasks such as Montezuma's Revenge. Figure 10, shows how the replay buffer viewport can be used to visualize the distribution of data samples in the replay buffer. Since, the observations returned from the environment are images, we extract the embeddings computed by the feature extraction model in the agent policy, and use these for the projection technique described in the Appendix (Algorithms).

### USER STUDY

We conducted a user study[6] from RL users for feedback and an extended evaluation of potential use cases a tool such as the one described and implemented in this paper would serve.

These questions were of both a numerical and subjective type. We now list both types along with preliminary results

### NUMERICAL QUESTIONS

- On a scale of 0-10, do you think Vizarel would help you identify bugs in your RL algorithms? **Average: 7.5**

- On a scale of 0-10, do you think Vizarel would help you identify improvements in your RL algorithms? **Average: 7.0**

- On a scale of 0-10, do you think Vizarel would help you understand whether your RL algorithms are working as intended? **Average: 8.0**

### SUBJECTIVE QUESTIONS (SELECTED)

- Are there specific settings where you think this tool might help answer questions that you might otherwise not easily been able to?
  - To show the effects of changes in reward function coefficients
  - Surfacing important points in the agent trajectory history

- Which features do you think are missing and would be a useful addition to have?
  - Add the capability to search over the replay buffer viewport and filter events based on search criteria.
  - Provide a documented approach to load in agent policies
  - More details on how to create plugins

---

[6]Results from this are preliminary, as the survey is still in progress

– Display agent trajectories in the replay buffer viewport

Future work along this direction would include creating test scenarios for debugging, and running an A/B test for users, contrasting their experience with existing tools vs the proposed tool along dimensions of efficacy in debugging RL algorithms.

PERFORMANCE

We've run measured preliminary performance metrics to help provide insight into how much overhead running this system would create[7]. These numbers were collected for a visualization of an agent trained using DDPG on the HalfCheetah-v2 task, about 35% of the way to task completion. The vizarel interface was generated on a machine with an Intel(R) Xeon(R) CPU E5-2620 v4 @ 2.10GHz processor, with a 4TB HDD, and 128 GB RAM.

| Tasks | Time |
|---|---|
| Generate Viewports | 45 sec |
| Load Dashboard | 5 sec |
| Policy Rollouts | 2 minutes |
| Logging Overhead Fraction (Relative to Tensorboard) | 1 |

Note that the policy rollout time is conditional on the length of the episode trajectory. These were collected for the HalfCheetah-v2 task

---

[7]Note that these numbers may not be indicative of the final numbers, since the codebase is under active development

