# OpenReview forum: "Interactive Visualization for Debugging RL"
_ICLR.cc/2021/Conference — Reject_

### Official Review · AnonReviewer1 · 2020-10-26
**A new interactive RL visualization tool is described. Needs more detailed use cases to demonstrate its value.**

**Rating:** 5
**Confidence:** 3

**Review:**

This paper describes a new interactive visualization tool for better debugging of RL algorithms. The tool provides two fundamental views, spatial and temporal views of the data generated by an RL algorithm. Various viewports under the two different views are described in details.

This tool is definitely useful to many RL researchers because debugging RL algorithms is known to be more difficult than traditional ML algorithms. The paper appears to have covered every component of the tool in details and a brief example of walkthrough. However, I feel that the paper has not yet sufficiently demonstrated the proposed tool’s full value. The tool provides a number of viewports, each covering a particular aspect of the RL algorithm under investigation. Is there any mechanism for the user to jointly operate several viewports (e.g., state, action, reward) together? What are the use cases that would be valuable to many users? The paper briefly describes one, but I think a better way is to offer multiple detailed case studies detailing the following points: 1. what is the problem being addressed (since we are debugging)? 2. How does the user operate through a combination of different viewports to find the source of the problem? This way, we will have a better idea on when the tool is most effective and how the different components contribute to the values created.

Minor comment: The citation (noa, 2020a) under 4.2.1 seems incorrect.

---

> ### Author Response · Authors · 2020-11-24
> **Response to Review — Thank you for the feedback!**
>
> > Is there any mechanism for the user to jointly operate several viewports (e.g., state, action, reward) together?
>
> This feature doesn’t exist yet, but wouldn’t be challenging to implement with the current state of the codebase.
>
> >What are the use cases that would be valuable to many users? The paper briefly describes one, but I think a better way is to offer multiple detailed case studies detailing the following points: 1. what is the problem being addressed (since we are debugging)? 2. How does the user operate through a combination of different viewports to find the source of the problem?
>
> We’ve added a section to the appendix that has preliminary results from a user study we are currently running. We will have more results from both the user study in the upcoming weeks.
>
> In addition, we’ve added a section that shows a visualization of using this tool on a representative hard exploration task (Montezuma’s Revenge). We plan to add more comprehensive examples of potential use cases for debugging and which viewports could be used to diagnose the problem, at the time of tool and documentation release.
>
> We’ve also added a section in the appendix on a basic set of performance measures. In addition, we’ve cited additional references in the related work (Section 2). We’ve also the suggested errors from the main text and in the references section.

---

### Official Review · AnonReviewer3 · 2020-10-26
**Potentially important problem with excellent demo, but anonymisation failure, lack of experimental validation, missing references, and unclear prose. Recommend rejection.**

**Rating:** 4
**Confidence:** 3

**Review:**

	1. Summary of Paper
		a. This article contributes a framework and tool for visualising data collected from a policy during RL training. It also contributes a set of views for visualising RL training data that could be used in general. Finally, the paper contributes a high-level workflow and a set of example use cases for how this tool might impact debugging a training session.
	2. Strengths
		a. Problem Motivation
			i. The visualisation process for debugging/understanding RL training is certainly different from supervised learning use case. There is a need for a general tool to perform debugging on RL-trained models. This could be partially satisfied using visualisation methods based on the samples collected during training.
		b. Clarity
			i. The article provides a supplementary link in the main manuscript to an anonymised demo of the tool on (potentially) simulated data. This really helps understand the use of the tool by interacting with it. This was particularly helpful to understand the descriptions in the static manuscript.
			ii. The article also used figures effectively providing a clear understanding of each type of viewport described.
		c. Technical Approach
			i. The usage of simple visual effects to guide the user's attention and "nudge" them is useful.
	3. Weaknesses
		a. Anonymisation Failure in References
			i. A reference uncited in the manuscript body contains a non-anonymised set of author names to a paper with the same title as the system presented in this paper. This was not detected during initial review. "Shuby Deshpande and Jeff Schneider. Vizarel: A System to Help Better Understand RL Agents.arXiv:2007.05577."
		b. Citations
			i. There is egregiously missing information in almost all citations. This is quite obvious in all the missing dates in the manuscript text.
		c. Clarity
			i. There are a few unclear or misleadingly worded statements made as below:
				1) "However, there is no corresponding set of tools for the reinforcement learning setting." - This is false. See references below (also some in the submitted paper).
				2) "stronger feedback loop between the researcher and the agent" - This is at least confusing. In any learning setting, there is a strong interaction loop between experimentation by the researcher and resulting outcomes for the trained model.
				3) "To the best of our knowledge, there do not exist visualization systems built for interpretable reinforcement learning that effectively address the broader goals we have identified" - It isn't clear what these broader goals are that have been identified. Therefore it isn't possible to evaluate this claim.
				4) "For multi-dimensional action spaces, the viewport could be repurposed to display the variance of the action distribution, plot different projections of the action distribution, or use more sophisticated techniques (Huber)." - It would be clearer to actually state what the sophisticated techniques from Huber are here.
			ii. The framework could be clearer that it applies most directly as described on single agent RL. The same approach could be used with multi-agent RL but the observation state and visualisations around that get more confusing when there are multiple potentially different sets of observations. Is this not the case, please clarify?
		d. Experimental rigour
			i. The paper does include 3 extremely brief examples of how the tool might be used. However, it does not include any experiments to suggest that this tool would actually improve the debugging process for training RL in a real user study. Not every paper requires a user study, however, the contributions proposed by this particular manuscript require validation at some level from actual RL users (even a case study with some feedback from users would address this to some extent).
		e. Novelty in Related Work
			i. The manuscript contains references to several relevant publications that can be compared to the current work. However, the paper is also missing references to many related and relevant works in the space of debugging reinforcement learning using visualisations, especially with an eye towards explainable reinforcement learning. See a sampling below.
				1) @inproceedings{ Rupprecht2020Finding, title={Finding and Visualizing Weaknesses of Deep Reinforcement Learning Agents}, author={Christian Rupprecht and Cyril Ibrahim and Christopher J. Pal}, booktitle={International Conference on Learning Representations}, year={2020}, url={https://openreview.net/forum?id=rylvYaNYDH} }
				2) @inproceedings{ Atrey2020Exploratory, title={Exploratory Not Explanatory: Counterfactual Analysis of Saliency Maps for Deep Reinforcement Learning}, author={Akanksha Atrey and Kaleigh Clary and David Jensen}, booktitle={International Conference on Learning Representations}, year={2020}, url={https://openreview.net/forum?id=rkl3m1BFDB} }
				3) @inproceedings{ Puri2020Explain, title={Explain Your Move: Understanding Agent Actions Using Specific and Relevant Feature Attribution}, author={Nikaash Puri and Sukriti Verma and Piyush Gupta and Dhruv Kayastha and Shripad Deshmukh and Balaji Krishnamurthy and Sameer Singh}, booktitle={International Conference on Learning Representations}, year={2020}, url={https://openreview.net/forum?id=SJgzLkBKPB} }
				4) @article{reddy2019learning, title={Learning human objectives by evaluating hypothetical behavior}, author={Reddy, Siddharth and Dragan, Anca D and Levine, Sergey and Legg, Shane and Leike, Jan}, journal={arXiv preprint arXiv:1912.05652}, year={2019} }
				5) @inproceedings{mcgregor2015facilitating, title={Facilitating testing and debugging of Markov Decision Processes with interactive visualization}, author={McGregor, Sean and Buckingham, Hailey and Dietterich, Thomas G and Houtman, Rachel and Montgomery, Claire and Metoyer, Ronald}, booktitle={2015 IEEE Symposium on Visual Languages and Human-Centric Computing (VL/HCC)}, pages={53--61}, year={2015}, organization={IEEE} }
				6) @article{puiutta2020explainable, title={Explainable Reinforcement Learning: A Survey}, author={Puiutta, Erika and Veith, Eric}, journal={arXiv preprint arXiv:2005.06247}, year={2020} }
				7) @book{calvaresi2019explainable, title={Explainable, Transparent Autonomous Agents and Multi-Agent Systems: First International Workshop, EXTRAAMAS 2019, Montreal, QC, Canada, May 13--14, 2019, Revised Selected Papers}, author={Calvaresi, Davide and Najjar, Amro and Schumacher, Michael and Fr{\"a}mling, Kary}, volume={11763}, year={2019}, publisher={Springer Nature} }
				8) @inproceedings{juozapaitis2019explainable, title={Explainable reinforcement learning via reward decomposition}, author={Juozapaitis, Zoe and Koul, Anurag and Fern, Alan and Erwig, Martin and Doshi-Velez, Finale}, booktitle={IJCAI/ECAI Workshop on Explainable Artificial Intelligence}, year={2019} }
				9) @misc{sundararajan2020shapley, title={The many Shapley values for model explanation}, author={Mukund Sundararajan and Amir Najmi}, year={2020}, eprint={1908.08474}, archivePrefix={arXiv}, primaryClass={cs.AI} }
				10) @misc{madumal2020distal, title={Distal Explanations for Model-free Explainable Reinforcement Learning}, author={Prashan Madumal and Tim Miller and Liz Sonenberg and Frank Vetere}, year={2020}, eprint={2001.10284}, archivePrefix={arXiv}, primaryClass={cs.AI} }
				11) @article{Sequeira_2020, title={Interestingness elements for explainable reinforcement learning: Understanding agents’ capabilities and limitations}, volume={288}, ISSN={0004-3702}, url={http://dx.doi.org/10.1016/j.artint.2020.103367}, DOI={10.1016/j.artint.2020.103367}, journal={Artificial Intelligence}, publisher={Elsevier BV}, author={Sequeira, Pedro and Gervasio, Melinda}, year={2020}, month={Nov}, pages={103367} }
				12) @article{Fukuchi_2017, title={Autonomous Self-Explanation of Behavior for Interactive Reinforcement Learning Agents}, ISBN={9781450351133}, url={http://dx.doi.org/10.1145/3125739.3125746}, DOI={10.1145/3125739.3125746}, journal={Proceedings of the 5th International Conference on Human Agent Interaction}, publisher={ACM}, author={Fukuchi, Yosuke and Osawa, Masahiko and Yamakawa, Hiroshi and Imai, Michita}, year={2017}, month={Oct} }
	4. Recommendation
		a. I recommend this paper for rejection as the degree of change needed to validate the contribution through a user study or other validation is likely not feasible in the time and space needed. However, the contribution is potentially valuable to RL, so with the inclusion of this missing evaluation and additions to related work/contextualisation of the contribution, I would consider increasing my score and changing my recommendation.
	5. Minor Comments/Suggestions
		a. It is recommended to use the TensorFlow whitepaper citation for TensorBoard (https://arxiv.org/abs/1603.04467). This is the official response (https://github.com/tensorflow/tensorboard/issues/3437).

---

> ### Author Response · Authors · 2020-11-24
> **Response to Review — Thank you for the feedback!**
>
> >1) "However, there is no corresponding set of tools for the reinforcement learning setting." - This is false. See references below (also some in the submitted paper).
>
> We agree with the evaluation and have reworded this sentence to reflect more nuance. Our views are that the contribution here is not on any single technique to increase interpretability, but a whole suite of visualizations, built on an extensible platform to help researchers better design and debug RL agent policies. We’ve also added the suggested references to the related work section.
>
> >2) "stronger feedback loop between the researcher and the agent" - This is at least confusing. In any learning setting, there is a strong interaction loop between experimentation by the researcher and resulting outcomes for the trained model.
>
> Yes, that’s a good observation. What we intended to claim here is that the interaction loop between researcher and agent is stronger and more direct in the case of RL compared to the supervised learning setting. The example offered of experimentation and observing resulting outcomes is true also in the case of RL but is one step “meta” to the original process. The process of an agent “interacting” with the environment and collecting data samples conditional on the quality of the current policy is more unique to the RL setup and could be more aptly compared to the active learning setting.
>
> We make this claim since we believe that there should exist tools specifically designed around this “interactive” framework.
>
> >3) "To the best of our knowledge, there do not exist visualization systems built for interpretable reinforcement learning that effectively address the broader goals we have identified" - It isn't clear what these broader goals are that have been identified. Therefore it isn't possible to evaluate this claim.
>
> Yes, this is a valid critique. The broader goals weren’t explicitly stated, however, we intended to convey that using existing tools it is harder to answer questions of the nature we’ve stated:
>
> * How does the agent state-visitation distribution change as training progresses?
> * What effect do noteworthy, influential states have on the policy?
> * Are there repetitive patterns across space and time that result in the observed agent behavior?
>
> > ii. The framework could be clearer that it applies most directly as described on single agent RL. The same approach could be used with multi-agent RL but the observation state and visualisations around that get more confusing when there are multiple potentially different sets of observations. Is this not the case, please clarify?
>
> Yes, this is the case. The current system is designed with single agent RL in mind. For multi-agent systems, two possible extensions are:
>
> 1. store additional metadata to associate logs with specific agent IDs and then filter / visualize these
> 2. spawn multiple vizarel sessions to keep track of agents based on IDs (if being logged)
>
> We’ve attempted to rectify and work on the other suggestions raised, and have also fixed numerous minor errors both in the main text and in the references.
>
> > The paper does include 3 extremely brief examples of how the tool might be used. However, it does not include any experiments to suggest that this tool would actually improve the debugging process for training RL in a real user study. Not every paper requires a user study, however, the contributions proposed by this particular manuscript require validation at some level from actual RL users (even a case study with some feedback from users would address this to some extent).
>
> We’ve added a section to the appendix that has preliminary results from a user study we are currently running. In addition, we’ve added a section that shows a visualization of using this tool on a representative hard exploration task (Montezuma’s Revenge). We will have more results from both the user study in the upcoming weeks, along with examples of more such use cases where this tool could be used. In addition to this, we’ve added a section on a basic set of performance measures.
>
> We have also added all of the suggested references to the related work (Section 2), made additional changes based on suggestions, and fixed the errors from the main text and in the references.

---

### Official Review · AnonReviewer2 · 2020-10-28
**The proposed interactive RL visualiation framework already provides insightful statistics of the environment and policy, but lacks design decisions and a thorough empirical evaluation.**

**Rating:** 3
**Confidence:** 4

**Review:**

The paper deals with debugging of black-box deep reinforcement learning (RL) agents to better understand and fix their policies. The authors propose diverse tools for, among others, visualizing the state space in terms of calculated statistics, analyzing the taken actions across learning episodes or exploring the replay buffer. The authors also propose a workflow for using the proposed tools. The resulting Vizarel tool is evaluated in terms of an exemplary walkthrough.

The approach follows an interesting direction towards explaining RL agents, but I am missing concrete design decisions and empirical evaluations for the proposed set of visualizations. While evaluating interpretability/explainability is difficult in general, it is still essential for such kind of contribution. I feel that the authors should explore some kind of user study (as conducted in cited works, such as contrastive RL explanations [1]), where end-users need to solve a challenging RL-related task and use the tools for actual debugging/search for improvements. I am aware that other explainable RL approaches based on counterfactuals or attentions might be easier to evaluate (and might not require end-users for acceptance at a conference), but I still feel a deeper evaluation is necessary here. To this end, an evaluation then should include mentioned related works on explainable RL in order to empirically prove the superiority for specific tasks / use cases.

To this end, I am wondering it which situations the tool would be beneficial over other explainable RL approaches. For example, does the approach work well for procedural, hard-exploration tasks? More specifically, which tools of the framework would I use and which potentially not? Again, I feel like such questions can only be answered by asking actual end-users.

Lastly, there seem to be numerous minor errors and in the references, as publication years are often missing.

[1] van der Waa, J., van Diggelen, J., Bosch, K.V.D. and Neerincx, M., 2018. Contrastive explanations for reinforcement learning in terms of expected consequences. arXiv preprint arXiv:1807.08706.

--- Update after author response period ---
Thank you for the clarifications! After reading the other reviews and the paper updates, I still feel that the paper requires an additional, empirical evaluation to prove the value and contribution of the approach. Otherwise I find it hard to tell to what extent / in which situations / for which user group the tool is useful. While I appreciate that the authors made changes to their manuscript based on the reviewers' comments, I thus keep my recommendation for rejection for the current version of the paper.

---

> ### Author Response · Authors · 2020-11-24
> **Response to Review — Thank you for the feedback!**
>
> We appreciate the suggestion of running a user study, and are currently running one. We’ve already received preliminary answers to questions that we’ve asked, which we’ve added to the appendix section. We will have more results from both the user study in upcoming weeks.
>
> Running a user study is something that we considered running as well prior to submitting the paper. We’re aware of existing work that proposed techniques for interpretable RL (van der Waa et al. (2018) for example), and ran a user study to gauge the effectiveness of such a visualization. However, we think the contribution here is not on any single technique to increase interpretability, but a whole suite of visualizations and an extensible platform that could help researchers better design and debug RL agent policies for their task. To that end, we’ve received feedback from many RL users echoing that such a tool would be highly useful in their debugging workflows, as well as suggestions for directions of potential improvement.
>
> We’ve added a section (Appendix) that shows a visualization of using this tool on a representative hard exploration task (Montezuma’s Revenge), and will be providing more examples of where this tool could be used at the time of code release.
>
> In the appendix, we've also added brief section that provides information about a basic set of performance measures.
>
> >For example, does the approach work well for procedural, hard-exploration tasks? More specifically, which tools of the framework would I use and which potentially not?
>
> Yes we think this approach would work well for procedural, hard exploration tasks since we provide a way to visualize and increase insight into the exploration process. We’ve added a section to the appendix visualizing the use of this tool on Montezuma’s Revenge (chosen as a representative example of a hard exploration task). We aim to provide an exhaustive set of components through this tool that the user could use. For components not available we aim to create an easy way for users to release their contributions through an ecosystem of “plugins”.
>
> Through our experience speaking with end users, we’ve realized that there is still much scope for improvement on which features should be included, however we believe that this is a process of iterative feedback and wouldn’t effectively take place until this tool is released and undergoes testing through a broad set of use cases.
>
> We’ve attempted to fix the numerous minor errors both in the main text, and in the references.

---

### Official Review · AnonReviewer4 · 2020-10-29

**Rating:** 6
**Confidence:** 5

**Review:**

This paper is introducing a tool intended for analysing RL processes based on visual information, in order to help the researchers to understand what is happening with the agent environment interaction. The system has varied options for plotting useful information that researchers/engineers normally have to analyze for debugging, but additionally it allows the user to have interaction data in the plots and visualizations of the environments.
- This kind of works are promising and have some valuable contribution in order to unify and standardize the tools in the community, so that the reproducibility is more straightforward, as it has been with some RL libraries, sets of environments, or even the libraries for deep learning. However, the paper is too much focused on the side of showing what can be visualized by the users interactively, and it is missing more technical information about it. I assume there will be manuals, readmes, or so, but some technical information about the implementation of the tool would make the paper more useful for the readers, otherwise, going through this paper will not be necessary at all once the repo of the tool is released along with its documentation (making the published paper only useful for citing the use of the tool).
- Does the tool bring a library of algorithms and environments? or is it built to be compatible with any implementation? If that is the case, what is the architecture of the system? which helps the users to understand how to connect their code with the tool. Without this information, it is not possible for the users to extend its use to other applications.
- Figures need to be more connected with text in the paragraphs, some are not mentioned and some are mentioned very far from its place.
- In temporal views it could be also possible to visualize value functions, advantage functions, returns, and also the cost functions computed for training the models (NNs).
- In Section 4.1.2 it is mentioned "This idea can easily be extended to agents with stochastic actions, where we could generate a viewport using histograms to visualize the change in action distribution over time", this indeed might be a better example rather than the one given in Figure 2.
- Also in Section 4.1.3  it is mentioned "The viewports discussed so far can be combined to provide the user more insight into the correspondence between states (stateviewport), actions (action viewport), and the components of the reward function (reward viewport)that the agent is attempting to maximize.  Such a visualization could help alert the researcher toreward hacking (Amodei et al.) and thus design reward functions that are immune to this problem.", a good example in a Figure would be very depicting.
- Can this tool be used to interactively change settings while running learning processes? e.g. graphically modifying reward functions, selecting regions of points in the replay buffer to be sampled more often or to be ignored, etc.
- In section 4.2.1 it is mentioned "We can instead visualize the data samples by transforming the points (van der Maaten & Hinton, 2008) to a lower-dimensional representation. This technique helps visualize the distribution of samples in the replaybuffer, which is a visual representation of the replay buffer diversity". It would be good to add the name of the technique, not only the reference.
- What does  "ask questions" means in Section 4.2.2, in the sentence: "The distribution viewport (Figure 4) complements the replay buffer viewport by allowing the user to select clusters of data samples and ask questions..."
- It would be good to add some comments about computational performance and limitations.
- Is it useful for doing experiments with real physical systems?
- Small detail, Figure 2 is not split in left and right.

---

> ### Author Response · Authors · 2020-11-24
> **Response to Review — Thank you for the feedback!**
>
> Yes, we agree with the observation that certain parts of the paper that explain the implementation of the tool might become redundant once the technical documentation is released. However, we believe that the paper in its current form is structured to provide context as to how this tool differs from existing tools (Section 1 & 2), along with comments with context about potential use cases in the remaining sections.
>
> The tool is designed to be compatible with popular implementations of algorithms and environments. For our own testing, we’ve been using stable-baselines3, but the API is agnostic to the specific choice of the library. We felt providing an architecture diagram wouldn’t provide as much insight into how the tool is used, and hence provided a workflow diagram instead. However, we are eager to release this along with the documentation if that would be helpful.
>
> We’ve expanded with more clarifications in the appendix as well about an ongoing user study, usage in alternate environments, and preliminary performance metrics.
>
> > Also in Section 4.1.3  it is mentioned "The viewports discussed so far can be combined to provide the user more insight into the correspondence between states (stateviewport), actions (action viewport), and the components of the reward function (reward viewport)that the agent is attempting to maximize.  Such a visualization could help alert the researcher to reward hacking (Amodei et al.) and thus design reward functions that are immune to this problem.", a good example in a Figure would be very depicting.
>
> Yes, we agree, these could be interesting examples to visualize!
>
> > Can this tool be used to interactively change settings while running learning processes? e.g. graphically modifying reward functions, selecting regions of points in the replay buffer to be sampled more often or to be ignored, etc.
>
> Yes, it could be. However, we made the design choice to not implement this functionality at the moment due to state synchronization issues when making changes to live running agents. We found that it was simpler to operate over policy checkpoints for the purpose of visualizations. That said, we are currently working on this feature.
>
> > What does  "ask questions" means in Section 4.2.2, in the sentence: "The distribution viewport (Figure 4) complements the replay buffer viewport by allowing the user to select clusters of data samples and ask questions..."
>
> Users might ask questions like:
> * What is the distribution of actions the agent took for these groups of similar states?
> * What is the distribution of rewards for the state action transitions?
> * What is the overall diversity of states which the agent has visited?
>
> We emphasize that Vizarel is not constrained to answering these specific questions, but rather can be used to facilitate interactive debugging. We have added this clarification to Section 4.2.2.
>
> > It would be good to add some comments about computational performance and limitations.
>
> We have added a section to the appendix on preliminary performance metrics. Beyond this, we are in the process of designing a more comprehensive set of measures that might be useful, which we plan to release along with the tool.
>
> > Is it useful for doing experiments with real physical systems?
>
> Yes, this definitely could be. However, one question would be whether the policy rollout mechanism for the physical system is robust enough, with safe auto reset capability. We have not tried this with an actual physical system, however, since all of our experiments have been with agents in simulated environments.

---

### Decision · Program_Chairs · 2021-01-07
**Final Decision**

**Decision:**

Reject

**Comment:**

We also had some discussions about the paper that are not visible to the authors. To summarize: the reviewers appreciated the efforts the authors put into the replies and updates. While those clarifies quite a few points, the paper unfortunately is still not publishable in its current form at ICLR.

Overall the paper tackles a very relevant and important question, proposing a tool that could be extremely useful for research on RL.
On the downside the paper is mainly descriptive, outlining WHAT the tool can do. Multiple reviewers pointed out that deeper, new insights are missing, e.g., WHY certain features were included and whether the tool actually is helpful for practitioners. A user study has been commenced, which is an excellent step in this direction.